# Quality of *Oreochromis niloticus* and *Cynoscion virescens* fillets and their by-products in flours make for inclusion in instant food products

Stefane Santos Corrêa[1]☯, Gislaine Gonçalves Oliveira[1]☯, Melina Coradini Franco[1]☯, Eliane Gasparino[1]☯, Andresa Carla Feihrmann[2]☯, Simone Siemer[1]☯, Jerônimo Vieira Dantas Filho[3,4]☯*, Jucilene Cavali[4]☯, Sandro de Vargas Schons[3]☯, Maria Luiza Rodrigues de Souza[1]☯

1 Programa de Pós-Graduação em Zootecnia, Universidade Estadual de Maringá (UEM), Maringá, Paraná, Brazil, 2 Departamento de Engenharia de Alimentos, Universidade Estadual de Maringá (UEM), Maringá, Paraná, Brazil, 3 Programa de Pós-Graduação em Ciências Ambientais, Universidade Federal de Rondônia (UNIR), Rolim de Moura, Rondônia, Brazil, 4 Programa de Pós-Graduação em Sanidade e Produção Animal Sustentável, Universidade Federal do Acre (UFAC), Rio Branco, Acre, Brazil

☯ These authors contributed equally to this work.
* jeronimovdantas@gmail.com

**Data Availability Statement:** All relevant data is within the paper.

## Abstract

The production of fish flour is an alternative for better use of the raw material, although it is rarely used in instant food. Thus, the aimed of this study was to evaluate *Oreochromis niloticus* (Nile tilapia) and *Cynoscion virescens* (croaker) fillets and the elaboration of flour with filleting by-products for inclusion in food products. Carcasses and heads of the two fish species were cooked, pressed, ground, subjected to drying and re-grinding to obtain standardized flours. These carcass flours were seasoned (sweet and salted). This study was organized into two experimental tests: Test 1: Yield, physicochemical and microbiological analyzes of fillets and flours made from carcass and head of Nile tilapia and croaker; Test 2: Seasoned flours made from Nile tilapia carcasses. There was a difference in fillets yield, where the croaker demonstrated 46.56% and the Nile tilapia 32.60%. Nile tilapia fillets had higher protein content (17.08%) and lower lipid content (0.89%) compared to croaker fillets (14.21 and 4.45%). Nile tilapia backbone flour had the highest protein content (55.41%) and the croaker the highest ash (45.55%) and the lowest Nile tilapia (28.38%). The head flours had lower protein contents (39.86%). Flours produced with croaker backbone had higher levels of calcium and phosphorus (9.34 and 9.27%). However, Nile tilapia backbone flour showed higher contents of essential amino acids. These flours demonstrated a fine granulometry (0.23 to 0.56 mm). Seasoned flours demonstrated interaction between fish species and flavors for moisture, ash, carbohydrates, calcium and phosphorus. The highest protein content (29.70%) was for Nile tilapia flour sweet flours (31.28%) had higher protein content, while salted lipids (8.06%). Nile tilapia has a lower fillet yield, although with a high protein content and low lipid content. Comparing the flours made from filleting by-products, the backbone flour has better nutritional quality, with Nile tilapia being superior to that of croaker, especially in terms of protein and amino acids.

**Funding:** CAPES/Brazil through Programa Nacional de Cooperação Acadêmica na Amazônia (PROCAD/ AM). Project approved 2018/L.1, in the postgraduate line in Biodiversity, production and animal health. All authors were equally awarded.

**Competing interests:** The authors have declared that no competing interests exist.

## Introduction

The growth of the world population causes a greater demand for food to meet the need for animal protein. With this increase, it consequently generates a greater amount of by-products and waste, although they can used as raw material for the preparation of other food products [1]. It is important to emphasize that in order to have sustainability in Aquaculture, the raw material must used to the fullest, as in other better consolidated animal production chains [2]. According to Caruso et al. [3], despite the constant growth of fish farming, extractive fishing is still the largest source of protein supply from fish. However, the decline of this activity due to overfishing is notable, not allowing the renewal of fish stocks, therefore, it must measures must established so that this decline does not occur, requiring better use of the raw material, increasing the amount of fish to consumed, even if it is through the elaboration of products from the by-products generated in the fish processing.

The demersal cyanid fish species are the most important in extractive fisheries, which are widely distributed along the Brazilian coast, constituting an important fishing resource for the country. They represent 22% marine landings and 9% continental landings. Among the species, the croaker (*Cynoscion virescens*, Cuvier, 1830) is very expressive, as it is large and has an elongated body, with a total length 10 to 200 cm [2, 3]. It is a species with a carnivorous feeding habit, occurring from west Atlantic to Southeastern coast of Brazil [4]. Fish farming in Brazil, in contrast to extractive fishing, has emerged as the fourth largest producer *Oreochromis niloticus* Linnaeus, 1758 (Nile tilapia) [5]. Nile tilapia is cultivated in all Brazilian states, even though there is no commercialization in all of them in the Northern region of Brazil [5]. This species was introduced in Brazil and has adapted well to environmental characteristics, shown rusticity and good development for fish farming [6].

The commercialization of fish is done mainly through the cut in fillet, this processing of fish to obtain fillet without the skin generates an average 60 to 72% of the raw material in residue [7]. It is important to compare the quality of fillets of two species, one of the most produced by fish farming and the other obtained by extractive fishing, regarding yield, chemical composition and waste generated, mainly addressing the quality of these in relation to the main product. One of the main strategies for using by-products from fish processing is through the production of flour for human and animal food, through the elaboration of savory and/or sweet ingredients to enrich instant portions, such as cup cakes, sachets, cookies, pâté, etc. [8, 9].

Flours produced with fish filleting residues can used for protein and mineral enrichment, to reduce the carbohydrate content and caloric value of various foods for human consumption, such as extruded snacks, lasagna, cereal bars, bread, cookies, etc. As seen before, several researches with fish flours for food enrichment are found, although seasoned flour would another commercial alternative, and perhaps with greater acceptance by the consumer for shown greater ease of application in several products, for reducing any possibility of residual flavor or odor of fish [10]. This fact often ends up interfering with the acceptance of the product depending on the level of its inclusion. With the need for a more conscious and environmentally correct use of by-products and waste from fish filleting, this fact makes food industries seek viable solutions for the recovery of by-products and waste, seeking new alternatives for their use in a more efficient way more efficient and healthier for consumers of fish products [9, 10].

In view of the overview presented, the aimed of this study was to characterize *O. niloticus* (Nile tilapia) and *C. virescen* (croaker) fillets and flours made from reuse of processing by-products of these species for human consumption, as well as to evaluate the nutritional, microbiological and sensory quality of these flours seasoned (sweet and salted).

## Material and methods

The study was conducted at Fish Technology Laboratory, at Experimental Farm Iguatemi (EFI), belongs to Universidade Estadual de Maringá (UEM). Data collection of measures and weights of *C. virescens* (croaker) were carried out at processing unit in Calçoene municipality, Amapá state coast, Brazil Data collected from *O. niloticus* (Nile tilapia) were collected at the slaughterhouse in Mandaguaçu municipality, Paraná state, Brazil.

The croaker used had an average body weight 2800 ± 200g, while the Nile tilapia 750 ± 60g. These were stunned on ice (croaker, on fishing boats) and the Nile tilapia in the processing unit. For the processing of both species, once in the slaugather units, the fillets and skins and other reserved residues were removed, except for the viscera that were discarded.

The fillets, heads and carcasses (backbone or vertebral column with the remaining filleting meat) of croaker were packaged and frozen (± 18˚C) to transported to EFI/UEM, where the experimental tests were carried out along with Nile tilapia. The transport of croaker by-products was carried out in isothermal boxes from Amapá state to FEI/UEM in Maringá, PR, by air. For Nile tilapia fillets and by-products, the same conservation and storage procedure was performed. All fillets and by-products, once identified, were stored in a freezer in the Fish Technology sector at EFI/UEM until the tests were carried out. Two tests were carried out, described below with the methodologies applied to perform them.

### Test 1. Yield, physicochemical and microbiological analysis of fillets and flours made from carcass and head of croaker and Nile tilapia

For this experimental test, they were 10 specimens of croaker and Nile tilapia used to evaluate the morphometric measurements, weights and processing yields (fillet and by-products). Analyzes were performed to characterize the fillets, such as chemical composition, fatty acid and amino acid profile, pH and water activity (aW). The analyzes performed were only descriptive.

At the EFI/UEM, the flours were prepared with by-products from processing of croaker and Nile tilapia, being used both for the heads and for carcasses or backbone (with the bones and the remaining meat from filleting). Both carcass and head flours followed the methodology described by Souza et al. [10], with modification. The raw materials were prepared (fins present in the ridges and gills of the heads were removed) and washed. Then, they were placed in a pressure cooker and added to the solution prepared in 20 liters of water, 1.0g of BHT antioxidant (butylhydroxy-toluene) and 2mL of peroxitane1512®. This volume was divided into four pressure cookers, making four repetitions for the production of backbone flour and then a new preparation for the four repetitions for head flour. The same procedure was performed for the two fish species Nile tilapia and croaker.

The raw materials were cooked for 60 minutes, starting the chronolgical count from pressure beginning. Once the pans were opened, the material was drained and pressed (capacity of 10 tons), in order to extract excess water and oil. The cake obtained from pressing was ground in an industrial meat grinder (model CAF-10, Brazil), with an 8 mm disc. Then, these were distributed in aluminum trays, weighed and placed in a forced air circulation oven (MA035/1, Marconi, Piracicaba, São Paulo state, Brazil) at 90˚C for 24 hours. This drying period was monitored for analysis of the dehydration process. After 24 hours, the material was ground again in a knife mill (Willye-model TE-650, Tecnal, Piracicaba, São Paulo state, Brazil), to obtain flour with fine granulometry. This same methodology was applied to preparation of Nile tilapia and croaker heads flour, according to the modified methodology of Souza et al. [10].

Flours obtained were properly packaged and stored in a freezer (H500, Electrolux, Brazil) at -18˚C until the moment of analysis and used in the preparation of seasoned flours (Test 2.) Only

a small portion of each treatment was kept in a refrigerator (TF39, Electrolux, Brazil) for microbiological analysis, which took place around 24 hours after the completion of its preparation.

## Test 2. Salted and sweet seasoned flours based on Nile tilapia and croaker carcasses

Due to the results obtained in Test 1, which revealed that flours made from the backbone of the two species of fish had higher protein content, it was decided that only these flours were seasoned, one being salted (Test 2). Seasoning and another sweet, following the methodologies described below. For salted flour, 200g (53.26%) of carcass flour (Nile tilapia = T1 and croaker = T2), 33g (8.79%) of sesame, 6g (1.60%) of sesame oil, 1g (0.27%) monosodium glutamate, 3g salted flour (0.80%), 10g (2.66%) nori, 7.5g (2%) fermented soy sauce, 1g (0. 27%) of oregano, 1g (0.27%) of dehydrated garlic, 20 g (5.33%) of dehydrated onion, 20 g (5.33%) of sun-dried tomato, 3g (0.8%) of smell green, 20g (5.33%) of chimichurri, 50g (13.32%) of flaxseed. Backbone flour of each fish species was mixed with sesame oil and fermented soy sauce in a container with low heat, where a homogeneous mass was obtained, later all the dehydrated ingredients mentioned above were added in the container, which remained on fire low for 10 minutes, always moving with a spoon, so as not to burn.

For the preparation of sweet flour were used 150g (30.24%) of backbone flour (Nile tilapia = T3 and croaker = T4), 150g of brown sugar (30.24%), 20g (4.03%) cinnamon, 50g (10.08%) 100% cocoa chocolate, 1g (0.20%) cloves, 15g (3.02%) grated ginger, 1g (0.20%) oregano, 9g (1.81%) of vanilla in 100g of water (20.16%). For preparation, oregano was added to water, due to its antioxidant properties and after three minutes of boiling, the *Arapaima gigas* (paiche) carcass flour and brown sugar were added. After the mass was well homogenized, the other ingredients were added to the process, leaving for 10 minutes on low heat, always stirring with a spoon. The mass obtained was placed in an oven at 60˚C (MA035/1, Marconi, Piracicaba, São Paulo state, Brazil) for 24 hours, then ground in a knife-type grinder (Willye-model TE-650, Tecnal, Piracicaba, São Paulo state, Brazil). After finishing the elaboration and cooling of the seasoned flours (salted and sweet), they were packaged and conditioned at a temperature of -18˚C (H500, Electrolux, Brazil) until the moment of analysis. A sample was separated for microbiological analysis.

The calculation of flour yield was performed as a function of the final product in relation to initial weight of the raw material. Regarding proximate composition analyzes were carried out at Laboratory of Food and Animal Nutrition, at UEM. For that, samples of the fillets (n = 10) and also 4 aliquots of each flour were used for determinations of proximate composition (moisture and ash) according to the methodology of the Association of Official Analytical Chemists AOAC. Crude protein contents were evaluated by semi-micro Kjeldahl method [11]. For the extraction of total lipids, the methodology described by Bligh and Dyer [12] was followed. Carbohydrate contents were estimated using a mathematical formula that considers the sum of the values of moisture, protein, lipids and ash, substituted by 100% [13]. While the total caloric value was obtained by the sum of the average values multiplication of protein, lipids and carbohydrates multiplied by the factors 4, 9 and 4, respectively. In relation to the flours dehydration curve was also determined the trays were weighed every 2 hours for 24 hours in which the flours were in a forced air oven at 60˚ C (MA035/1, Marconi, Piracicaba, São Paulo state, Brazil).

Regarding fatty acid and amino acid profile, from content of the lipid extraction determined by Bligh and Dyer method [12], the determination of the fatty acid profile was carried out through the transesterification of the lipid by-products of the flours, according to ISO methodology [14]. This procedure allows the methyl esters to separated and identified in a 14-A gas chromatograph (Shimadzu, Japan), equipped with a flame ionization detector and a fused silica

capillary column (100m x 0.25 mm di x 0.25 μm, CP-Sil 88). The peak areas (percentages of relative areas) are integrated by a CG-300 integrator-processor (and other scientific instruments).

Fatty acid identifications were carried out by the following criteria: comparison of retention times of methyl esters of sigma standards (USA) with those of samples and comparison of ECL (Equivalent Chain Length) values of methyl esters of samples with values from literature [15]. For characterization of the amino acid profile, it was performed according to methodology described by Hagen et al. [16], at CBO analysis Laboratory, in the Campinas city, SP, Brazil.

Analysis of mineral determination (calcium and phosphorus) was performed according to AOAC (2005) methodology [17]. And, to obtain pH of the croaker and Nile tilapia flour, they were homogenized with distilled water and submitted to a pH meter reading (DM 22, digimed). The determination of water activity was performed using the Labswift-Novasina device.

The flours were submitted to an analysis to evaluate the granulometry according to methodology of Kelte-Filho et al. [18]. In this procedure, a fraction of 100g is used, placed in a system of overlapping sieves (30, 35 and 50 mesh, respectively 600, 500 and 300 μm, according to [17] and these are subjected to agitation by time around 3 minutes, to check the amount of material retained in each sieve. To determine the geometric diameter, the method for determining granulometry of ingredients for use in swine and poultry rations was used, according to Vukmirović et al. [19].

For colorimetry, a colorimeter (MINOLTA CR-10, Minolta Camera Co, Osaka, Japan) was used, which defines the luminosity and the chroma a* and chroma b*. Luminosity is defined by L* which evaluates an extension from 0 (zero) which is considered the color black to 100 which means the color white, a* (red-green component) and b* (yellow-blue component) [20].

The microbiological analyzes performed on the flours were: most probable number (MPN) of coliforms at 35 and 40° C, *Staphylococcus coagulase positive* count in CFU gram[-1] and *Salmonella* spp. [21, 22].

## Statistical design

For Test 1, morphometric measurements, weights and processing yields (fillet and by-products), chemical composition, pH and aW were used one-way ANOVA, with two treatments (fish species–Nile tilapia and croaker) and 10 repetitions per treatment (n = 10). The experimental unit was the fish of each species. They were ANOVA two-way was used to analyze the flours, with two species of fish (Nile tilapia and croaker) and two types of by-products (backbone and head), with 4 replications.

For Test 2, they were ANOVA two-way was also used, with two species of fish (Nile tilapia and croaker) and two flavors of flour (salted-seasoning and sweet), with 4 replications.

The averages of the parameters analyzed in two assays were compared by the Tukey's test and Student's t test, considering a 5% probability, using the Statistical Analysis System Software (SAS Inst. Inc. Cary, NC, USA) (2010) [23]. The fatty acid profile averages are showed for Nile tilapia and croaker carcass and head flour. However, for the aminogram, only the averages of the amino acids found in Nile tilapia and croaker carcass flour are showed for their characterization.

## Results

### Test 1. Yield, physicochemical and microbiological analysis of fillets and flours made from carcass and head of croaker and Nile tilapia

**Fillets yield.** The croaker used for filleting had significantly higher body weight (2820g) compared to Nile tilapia (750g). There was a significant difference in fillet yield between the

evaluated species, with croaker shown a higher fillet yield 46.56%, while Nile tilapia 32.60%, generating 51.83 and 66.66% of filleting by-products, respectively (Table 1).

**Proximate composition of fillets and head and backbone flours.** Regarding proximate composition in Nile tilapia and croaker fillets, there were no differences (P>0.05) for moisture and ash contents, with average contents 81.56 and 1.07%, respectively. However, for crude protein and total lipids there were differences (P<0.05) between species. Nile tilapia had a higher crude protein content 17.08%, while croaker had a higher total lipid content 4.45% (Table 2). Flours prepared using Nile tilapia and croaker heads and backbones did not demonstrated an interaction effect, between the species and the parts of the fish used, for the moisture and lipid contents. However, when evaluating fish part alone, there was a difference for lipid content, where the flours produced from heads of the animals obtained a higher content 5.09%, when compared with flours made from backbone 4.10% (Table 2).

For protein and ash content, there was an interaction between the species and part of the fish used, as well as for the amount of calcium and phosphorus present in the four flours prepared. The treatment using Nile tilapia carcass had the highest protein content 55.41%, while the treatment using croaker backbone had the highest ash content 45.55%.

**Minerals, aW and pH of flours.** Minerals calcium and phosphorus, both were present in higher concentrations in the flour produced with 9.34 and 9.27% of croaker backbone, respectively (Table 2). In addition to low moisture content in the flours, there was no interaction effect or significant differences between the fish species and fish parts used, both for water activity (aW) and for hydrogenic potential (pH). The fish flours had an average water activity 0.19 and an average pH 7.06.

**Amino acid profile of flours and backbone.** According to amino acid profile in backbone flours of two fish species evaluated, the flour made from Nile tilapia had the highest levels for all amino acids, when evaluated separately, compared to flour made from croaker backbone (Table 3).

**Dehydration and colorimetry of flours produced from blackbone and head.** With measurement of reduction masses (kg) of materials during dehydration process, it was observed that the optimal time for the dehydration of the flours obtained from fish carcasses was lower than those proposed by the studies carried out (Fig 1).

For colorimetry, there was an interaction effect between the species and parts of the fish used. For parameters of lightness (L) and chromaticity a* (red-green component). While for

**Table 1. Yield and chemical composition of Nile tilapia and croaker fillets.**

| Yield (%) | Nile tilapia | Croaker | P value | C.V.[1] |
|---|---|---|---|---|
| Fillet | 32.60 ± 6.53[b] | 46.56 ± 6.53[a] | <0.0001 | 11.15 |
| By-products | 66.66 ± 6.92[a] | 51.83 ± 7.03[b] | <0.0001 | 8.34 |
| Live animal mass at slaughter (Kg) | 0.75 ± 0.97[b] | 2.82 ± 1.24[a] | <0.0001 | 13.89 |
| Fillet chemical composition (%) | | | | |
| Moisture | 80.24 ± 1.61[a] | 82.87 ± 1.58[a] | 0.0658 | 1.27 |
| Crude protein | 17.08 ± 1.44[a] | 14.21 ± 1.52[b] | 0.0118 | 5.13 |
| Ash | 1.04 ± 0.03[a] | 1.09 ± 0.02[a] | 0.8408 | 1.05 |
| Total lipids | 0.89 ± 1.78[b] | 4.45 ± 1.60[a] | 0.0003 | 13.73 |
| Caloric value (Kcal per 100g) | 76.36 ± 10.27[b] | 96.89 ± 8.76[a] | 0.0138 | 6.92 |

Averages on the line followed by the same letter are equal by Student's t test (P<0.05);

[1]C.V. = Coefficient of variation.

**Table 2. Chemical composition, minerals, pH and water activity (aW) of Nile tilapia and croaker flours produced from different parts of this fishes.**

| Flours | | Chemical composition | | | | | | | |
|---|---|---|---|---|---|---|---|---|---|
| | | Moisture | Crude protein | Ash | Total lipids | Calcium | Phosphorus | pH | aW |
| Nile tilapia | Backbone | 3.53 ± 0.12 | 55.41 ± 1.28[a] | 28.38 ± 0.77[c] | 4.28 ± 0.40 | 5.01 ± 0.03[c] | 5.47 ± 0.04[c] | 7.08 ± 0.04 | 0.18 ± 0.01 |
| | C Head | 3.18 ± 0.07 | 39.89 ± 3.25[c] | 36.85 ± 0.25[b] | 5.36 ± 0.62 | 6.17 ± 0.61[bc] | 7.86 ± 0.08[b] | 7.00 ± 0.02 | 0.19 ± 0.01 |
| Croaker | Backbone | 3.95 ± 0.12 | 43.75 ± 0.97[b] | 45.55 ± 0.29[a] | 3.92 ± 1.10 | 9.34 ± 0.03[a] | 9.27 ± 0.43[a] | 7.09 ± 0.01 | 0.23 ± 0.05 |
| | Head | 3.95 ± 0.12 | 39.83 ± 0.78[c] | 39.76 ± 5.87[b] | 4.84 ± 0.33 | 7.38 ± 0.02[b] | 8.11 ± 0.09[b] | 7.09 ± 0.01 | 0.23 ± 0.05 |
| Main effects | | | | | | | | | |
| Species | Nile tilapia | 3.35 ± 0.21 | 47.65 ± 8.79 | 32.62 ± 4.67 | 4.82 ± 0.72 | 5.56 ± 0.72 | 6.66 ± 1.31 | 7.04 ± 0.05 | 0.19 ± 0.01 |
| | Croaker | 3.50 ± 0.70 | 41.79 ± 2.28 | 42.65 ± 4.89 | 4.38 ± 0.88 | 8.86 ± 1.62 | 9.19 ± 1.21 | 7.05 ± 0.04 | 0.19 ± 0.01 |
| Parts of this fishes | Backbone | 3.74 ± 0.25 | 49.58 ± 6.47 | 36.97 ± 9.42 | 4.10 ± 0.77[b] | 7.67 ± 2.92 | 7.87 ± 2.64 | 7.08 ± 0.03 | 0.21 ± 0.04 |
| | Head | 3.11 ± 0.47 | 39.86 ± 2.11 | 38.30 ± 4.04 | 5.09 ± 0.53[a] | 6.75 ± 0.79 | 7.99 ± 0.16 | 7.00 ± 0.02 | 0.18 ± 0.02 |
| Probabilities | | | | | | | | | |
| Species (S) | | 0.5150 | 0.0006 | 0.0004 | 0.2961 | <0.0001 | <0.0001 | 0.3922 | <0.0001 |
| Parts of this fishes (P) | | 0.2213 | <0.0001 | 0.4569 | 0.0359 | 0.0008 | 0.4446 | 0.2220 | <0.0001 |
| Interaction S x P | | 0.2475 | <0.0001 | 0.0031 | 0.8414 | <0.0001 | 0.4755 | 0.1201 | 0.8457 |
| C.V.[1] | | 11.01 | 4.15 | 7.88 | 14.87 | 4.25 | 4.25 | 0.41 | 14.14 |

Averages in the same column followed by standard deviation with different lowercase letters are different from each other by Tukey's test and Student's t test. (P<0.05);
[1]C.V. = coefficient of variation.

the chromaticity b* (yellow-blue component) there was no interaction, nor differences between the species or parts of the fish used, whose average value was 10.04 (**Table 4**).

The closer to 100 value of luminosity, the lighter flour, in view of this, the treatments with the croaker species, using both the ridge 76.72 and the head 74.10, demonstrated highest luminosity. However, the treatment with Nile tilapia head flour showed lower luminosity 56.56, that is, in this treatment the flours were darker when compared to others (**Table 3**). Everything indicates that it is also associated with the lipid content, as this content was higher (5.35%) in this flour (**Table 2**). Meanwhile, the croaker had lowest lipid content and highest luminosity in the flour.

Chromaticity a* indicates the tendency of the flours to red-green colors, this parameter when related to luminosity showed an inversion. Flour with the highest tendency to red pigmentation was precisely Nile tilapia head flour that expressed lowest luminosity and highest lipid content, whose value was 1.82 for chroma a*. Note that chroma a* of Nile tilapia head and backbone flour showed no statistical difference (**Table 4**).

**Granulometry analysis.** Regarding granulometry of the flours prepared, there was interaction between the species and parts of the fish used (backbone and head). Croaker flours, from both backbone and head, had the smallest mean geometric diameters (DGM), 0.26 and 0.23 mm, respectively (**Table 3**). Quantitatively, Nile tilapia head flour had a higher lipid content 5.36% (**Table 2**), which explains the higher DGM demonstrated by this treatment.

## Test 2. Salted and sweet seasoned flours based on Nile tilapia and croaker carcasses

**Proximal composition of seasoned flours.** Seasoned flours showed an interaction between the species (Nile tilapia and croaker), and flavors (salted, seasoning and sweet) for moisture, ash and carbohydrates. However, the levels of protein and lipids showed no interaction between the two factors analyzed. However, for protein content there were significant differences between the species and flavors, when analyzed separately, while for lipid content

**Table 3. Amino acid (aas) profile of Nile tilapia and croaker flours.**

| Amino acids (%) | Backbone flours | |
|---|---|---|
| | Nile tilapia | Croaker |
| Phenylalanine* | 2.71 | 1.61 |
| Histidine* | 1.47 | 0.82 |
| Leucine* | 4.76 | 2.71 |
| Isoleucine* | 2.97 | 1.65 |
| Lysine* | 5.31 | 3.12 |
| Methionine* | 1.96 | 1.26 |
| Threonine* | 3.06 | 1.88 |
| Tryptophan* | 0.51 | 0.27 |
| Valine* | 3.12 | 1.86 |
| Aspartic acid | 4.52 | 1.97 |
| Glutamic acid | 8.50 | 4.56 |
| Alanine | 4.10 | 3.48 |
| Arginine | 4.16 | 3.16 |
| Cystine | 0.75 | 0.65 |
| Glycine | 4.91 | 5.86 |
| Proline | 3.47 | 3.42 |
| Serine | 2.56 | 1.53 |
| Taurine | 0.13 | 0.08 |
| Tyrosine | 2.28 | 1.30 |
| Total essential aas* | 25.87 | 15.18 |
| Total non-essential aas | 35.38 | 26.01 |
| Total aas (%) | 61.25 | 41.19 |
| Crude protein (%) | 62.16 | 46.12 |

*aas = essential amino acids.

there was a significant difference for the flavors of the flours when analyzed separately (**Table 5**).

In proximate composition, there was interaction for moisture, ash and carbohydrates, when considering four flours prepared. When analyzing flavor of these flours, the sweets of two species, Nile tilapia 6.08% and croaker 4.26% had a lower moisture content, while salted ones had a higher moisture content, despite the fact that salted tilapia flour had a significantly higher moisture content moisture content higher than all analyzed flours.

When analyzing the species for production of flour, Nile tilapia had higher protein content 29.70% compared to croaker 26.28%. When analyzing sweet flours, they had significantly higher protein content 31.28% than salted flours 24.70%. With the flavoring of the flour, there was a reduction in the protein content, since in **Table 2**, it appears that Nile tilapia demonstrated 55.41 and 43.75% of croaker. Therefore, there was a reduction in the protein content of standard flours for seasoned ones (salted and sweet), being 46.40% for Nile tilapia and 39.93% for croaker.

**Minerals analysis.** For ash and mineral content (calcium and phosphorus) there was an interaction and when the unfolding of these interactions was carried out, it was observed that the flours produced based on the sweet croaker backbone flour was significantly higher 27.68% than the other flours with salt Nile tilapia backbone having lowest content 17.82%. Likewise, there was an interaction for minerals, calcium and phosphorus in the seasoned

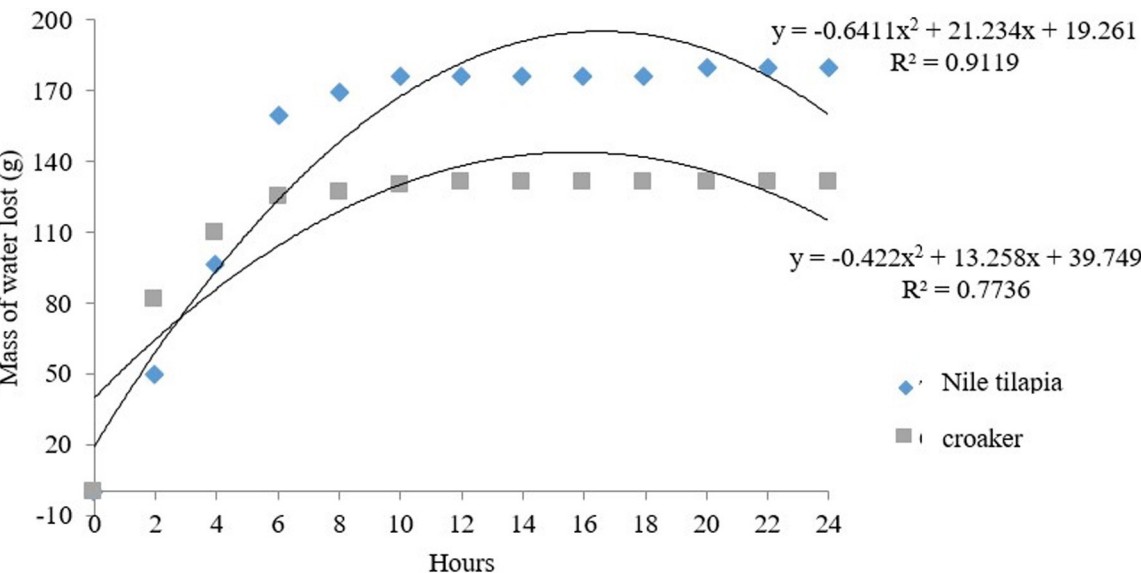

**Fig 1. Reduction of Nile tilapia and croaker flours masses during dehydration period.**

flours, with salt Nile tilapia flour having significantly lowest levels of calcium and phosphorus, while sweet croaker had the highest levels (**Table 5**).

**pH and aW of seasoned flours.** The fish specie did not influence the pH value of the elaborated flours, whose values were 7.06 Nile tilapia and 7.02 croaker. However, when these flours were seasoned, the sweet (7.71) had a higher pH value than the salty (6.37) (**Table 5**). When analyzing aW of the flours, it can seen in **Table 5** that there was an interaction for species and flavoring of the flours. Salted Nile tilapia flour had significantly higher aW 0.44 and sweet Nile

**Table 4. Colorimetry and granulometry of flours produced from backbone and head of Nile tilapia and croaker.**

| Flours | | Colorimetry | | | Granulometry (mm) |
|---|---|---|---|---|---|
| | | $L^*$ | $a^*$ | $b^*$ | |
| Nile tilapia | Backbone | $66.88 \pm 1.18^{ab}$ | $1.46 \pm 0.16^{ab}$ | $10.59 \pm 0.29$ | $0.36 \pm 0.02^{ab}$ |
| | C Head | $56.56 \pm 0.46^{b}$ | $1.82 \pm 0.12^{a}$ | $8.80 \pm 0.49$ | $0.56 \pm 0.03^{a}$ |
| Croaker | Backbone | $76.72 \pm 0.51^{a}$ | $0.63 \pm 0.08^{b}$ | $11.43 \pm 0.12$ | $0.26 \pm 0.02^{b}$ |
| | Head | $74.10 \pm 0.65^{a}$ | $0.21 \pm 0.07^{c}$ | $9.93 \pm 0.32$ | $0.23 \pm 0.03^{b}$ |
| Main effects | | | | | |
| Species | Nile tilapia | $61.72 \pm 5.71$ | $1.64 \pm 0.23$ | $9.10 \pm 1.05^{b}$ | $0.46 \pm 0.11$ |
| | Croaker | $75.42 \pm 1.52$ | $0.42 \pm 0.24$ | $10.68 \pm 0.85^{a}$ | $0.24 \pm 0.03$ |
| Parts of this fishes | Backbone | $71.80 \pm 5.45$ | $1.04 \pm 0.46$ | $11.01 \pm 0.50^{a}$ | $0.31 \pm 0.06$ |
| | Head | $65.33 \pm 9.62$ | $1.01 \pm 0.88$ | $9.37 \pm 0.72^{b}$ | $0.39 \pm 0.17$ |
| Probabilities | | | | | |
| Species (S) | | <0.0001 | <0.00001 | 0.0009 | <0.0001 |
| Parts of this fishes (P) | | <0,0001 | 0.6280 | <0.0001 | 0.0010 |
| Interaction S x P | | <0.0001 | 0.0003 | 0.4682 | 0.0002 |
| C.V.[1] | | 1.10 | 11.14 | 3.27 | 8.41 |

Averages in the same column followed by standard deviation with different lowercase letters are different from each other by Tukey's test and Student's t test. (P<0.05);
[1]C.V. = coefficient of variation.

**Table 5. Chemical composition, minerals, pH and water activity (aW) of Nile tilapia and croaker backbone seasoned flours.**

| Flours | | Chemical composition | | | | | | | | |
|---|---|---|---|---|---|---|---|---|---|---|
| | | Moisture | Crude protein | Total lipids | Ash | Carbohydrates | Calcium | Phosphorus | pH | aW |
| Nile tilapia | Salted | 6.08 ± 0.12[a] | 26.58 ± 2.20[a] | 7.72 ± 1.06[a] | 17.82 ± 0.29[c] | 41.80 ± 2.83[b] | 1.02 ± 0.01[c] | 1.79 ± 0.03[c] | 6.39 ± 0.02[a] | 0.44 ± 0.01[a] |
| | C **Sweet** | 4.17 ± 0.02[c] | 32.82 ± 0.67[a] | 3.70 ± 0.37[b] | 12.86 ± 0.08[d] | 46.44 ± 1.12[a] | 0.86 ± 0.03[d] | 1.22 ± 0.12[d] | 7.73 ± 0.02[a] | 0.22 ± 0.03[d] |
| Croaker | Salted | 5.42 ± 0.45[b] | 22.82 ± 1.06[a] | 8.41 ± 1.11[a] | 22.11 ± 0.36[b] | 41.23 ± 0.49[b] | 1.75 ± 0.03[b] | 2.53 ± 0.07[b] | 6.35 ± 0.04[a] | 0.35 ± 0.02[b] |
| | C **Sweet** | 4.26 ± 0.10[c] | 29.74 ± 1.47 | 4.13 ± 0.71[b] | 27.68 ± 0.09[a] | 34.18 ± 2.12[c] | 3.50 ± 0.02 | 3.99 ± 0.11[a] | 7.70 ± 0.01 | 0.29 ± 0.0 |
| Main effects | | | | | | | | | | |
| Species | Nile tilapia | 5.12 ± 1.04 | 29.70 ± 3.71[a] | 5.71 ± 3.19 | 15.34 ± 2.72 | 44.12 ± 3.19 | 0.94 ± 0.09 | 1.50 ± 0.32 | 7.06 ± 0.73 | 0.29 ± 0.07[c] |
| | Croaker | 4.84 ± 0.70 | 26.28 ± 3.96[b] | 6.27 ± 2.48 | 24.89 ± 3.05 | 37.70 ± 4.09 | 2.63 ± 0.96 | 3.76 ± 1.35 | 7.06 ± 0.73 | 0.33 ± 0.12 |
| Sabiruzada | Salted | 5.75 ± 0.46 | 24.70 ± 2.57[b] | 8.06 ± 0.55[a] | 19.96 ± 2.37 | 41.51 ± 1.84 | 1.39 ± 0.40 | 2.16 ± 0.41 | 6.37 ± 0.04[b] | 0.32 ± 0.03 |
| | **Sweet** | 4.22 ± 0.08 | 31.28 ± 1.98[a] | 3.92 ± 1.04[b] | 20.27 ± 8.12 | 40.31 ± 6.88 | 2.18 ± 1.45 | 3.11 ± 2.07 | 7.71 ± 0.02[a] | 0.40 ± 0.05 |
| Probabilities | | | | | | | | | | |
| Species (S) | | 0.0724 | 0.0037 | 0.2982 | <0.0001 | 0.0004 | <0.0001 | <0.0001 | 0.0656 | 0.25 ± 0.03 |
| Saborizada (Sb) | | 0.0724 | <0.0001 | <0.0001 | <0.0001 | 0.2999 | <0.0001 | <0.0001 | 0.0656 | 0.0007 |
| Interaction S x SB | | 0.0272 | 0.6985 | 0.8016 | <0.0001 | 0.0006 | <0.0001 | <0.0001 | 0.7684 | <0.0001 |
| C.V.[1] | | 4.79 | 5.24 | 14.47 | 1,19 | 4.58 | 4.58 | 4.58 | 0.40 | 1.40 |

Averages in the same column followed by standard deviation with different lowercase letters are different from each other by Tukey's test and Student's t test. (P<0.05);
[1]C.V. = coefficient of variation.

tilapia flour had lower aW 0.22. If compared within the same species, for croaker it occurred in a similar way, with the sweet smallest aW 0.29 and the salted largest aW 0.35.

**Microbiological analysis.** Seasoned flours also showed low coliforms at 35° and 45° C ([1]MPN g[-1]), being less than 3, for the *Staphylococcus coagulase positive* count (CFU g[-1]) it was also low, being less than 1x10[2] and was absent for *Salmonella* spp. in 25g.

**Fatty acids profile of seasoned flours.** The fatty acid profile of 8 treatments was evaluated, and 25 fatty acids were found in Nile tilapia carcass flour, 22 acids in Nile tilapia head flour, 23 acids in sweet Nile tilapia flour, 24 acids in salted Nile tilapia flour, 23 acids in croaker carcass flour, 23 acids in croaker head flour, 21 acids in sweet croaker flour and 22 acids in salted croaker flour (**Table 6**).

Among the acids found for croaker and Nile tilapia flours, Palmitic, Stearic, Palmitoleic, Oleic and Linoleic acids stand out, showes in different amounts (P<0.05) between treatments. Interestingly, Sapienic acid did not appear in treatments containing Nile tilapia flour and appeared in all treatments containing croaker flour, even in small amounts. While the dihomo-alpha-Linolenic, gamma-Linolenic and conjugated Linoleic acids did not appear in treatments containing croaker flour, showes a difference between the fatty acids expressed in the fish species, which may justify that croaker is an estuarine fish of saltwater and Nile tilapia fish from freshwater rivers and lakes. In relation to saturated fatty acids, the highest values were for croaker head flour 49.55 and for croaker carcass flour 49.31, fatty acids had the highest amount in salted flour, being 79.08 for salted Nile tilapia flour and 78.01 for salted croaker flour.

Salted flours of the two species had higher levels of omegas 3.6 and 9. However, the carcass and head of Nile tilapia flour, 36.34 and 34.57 respectively, had higher levels of omega 9, while the carcass and head 4.23 and 4.13 showed higher omega 3 content when compared to two species. In this study, the omega 6/omega 3 ratio for Nile tilapia flour was 13.58 for carcass, 17.98 for head, 13.90 for sweet and 1.91 for salt. While the omega 6/omega 3 ratio of croaker flours showed much lower values, being 1.12 for carcass, 1.08 for head, 2.25 for sweet and 1.66 for salt.

**Table 6. Fatty acid profile of flours made from Nile tilapia and croaker carcass, without and with added flavor.**

| Fatty acid (%) | Nile tilapia flour | | | | Croaker flour | | | |
|---|---|---|---|---|---|---|---|---|
| Usual nomenclature / symbology | Carcass | Head | Sweet | Salted | Carcass | Head | Sweet | Salted |
| Lauric acid[1]/ C12:0 | 0.09 | 0.07 | 0.13 | 0.04 | 0.16 | 0.15 | 0.09 | 0.06 |
| Myristic acid [1]/ C14:0 | 2.87 | 3.10 | 1.68 | 0.62 | 4.03 | 3.89 | 1.98 | 0.66 |
| Pentadecylic acid[1]/ C15:0 | 0.29 | 0.30 | 0.17 | 0.06 | 0.91 | 0.91 | 0.46 | 0.14 |
| Palmitic acid[1]/ C16:0 | 23.28 | 24.04 | 23.71 | 11.98 | 29.00 | 29.27 | 26.94 | 12.27 |
| Margaric acid[1]/ C17:0 | 0.42 | 0.43 | 0.31 | 0.12 | 1.26 | 1.29 | 0.76 | 0.22 |
| Stearic acid[1]/ C18:0 | 7.65 | 8.14 | 20.05 | 6.28 | 11.06 | 11.11 | 23.09 | 6.45 |
| Arachidic acid [1]/ C20:0 | 0.29 | 0.29 | 0.69 | 0.53 | 0.98 | 1.01 | 1.06 | 0.62 |
| Behenic acid [1]/ C22:0 | 0.14 | 0.11 | 0.21 | 0.25 | 1.03 | 1.04 | 0.56 | 0.35 |
| Lignoceric acid[1]/ C24:0 | 0.17 | 0.13 | 0.14 | 0.13 | 0.88 | 0.88 | 0.51 | 0.26 |
| Palmitoleic acid[2]/ C16:1 ω7 | 5.53 | 5.44 | 3.04 | 1.09 | 11.36 | 11.33 | 5.68 | 1.60 |
| Sapienic acid[2]/ C16:1 ω10 | - | - | - | - | 0.34 | 0.34 | 0.17 | 0.06 |
| Cis-10-heptadecenoic acid[2]/ C17:1 | 0.37 | 0.40 | 0.20 | 0.10 | 0.55 | 0.58 | 0.26 | 0.10 |
| Oleic acid[2]/ C18:1 ω9 | 32.53 | 34.21 | 33.43 | 31.27 | 14.96 | 14.63 | 25.52 | 28.85 |
| Vaccenic acid[2]/ C18:1 ω7 | 2.78 | 3.54 | 2.01 | 1.35 | 4.57 | 4.57 | 2.36 | 1.42 |
| Gondoic acid[2]/ C20:1 ω9 | 1.88 | 2.13 | 1.08 | 0.50 | 0.92 | 0.90 | 0.45 | 0.31 |
| Erucic acid[2]/ C22:1 ω9 | 0.16 | - | 0.07 | 0.13 | 0.22 | 0.20 | - | 0.16 |
| α-Linolenic acid[2]/ C18:3 ω3 | 0.70 | 0.54 | 0.52 | 15.20 | 0.31 | 0.29 | 0.31 | 16.48 |
| Stearidonic acid[2]/ C18:4 ω3 | 0.06 | - | - | - | 0.21 | 0.20 | 0.09 | 0.02 |
| Dihomo-α-linolenic acid[2]/ C20:3 ω3 | 0.22 | 0.13 | 0.11 | 0.03 | - | - | - | - |
| Eicosapentaenoic acid—EPA[2]/ C20:5 ω3 | 0.10 | - | 0.06 | 0.09 | 3.71 | 3.64 | 1.66 | 0.60 |
| Linoleic acid[2]/ C18:2 ω6 | 10.52 | 9.56 | 7.80 | 28.62 | 1.46 | 1.26 | 3.19 | 27.92 |
| Gamma linolenic acid—GLA[2]/ C18:3 ω6 | 0.81 | 0.63 | 0.45 | 0.14 | - | - | - | - |
| Conjugated linoleic acid—CLA[2]/ C18:2 ω6 | 0.20 | 0.13 | 0.11 | 0.03 | - | - | - | - |
| Eicosadienoic acid[2]/ C20:2 ω6 | 0.68 | 0.61 | 0.35 | 0.14 | 0.28 | 0.26 | 0.16 | 0.09 |
| Dihomo-Gamma-linolenic acid—DGLA[2]/ C20:3 ω6 | 0.77 | 0.51 | - | 0.14 | 0.22 | 0.22 | - | - |
| Arachidonic acid[2]/ C20:4 ω6 | 1.69 | 0.61 | 0.88 | 0.25 | 2.79 | 2.72 | 1.28 | 0.40 |
| Others* | 5.91 | 4.95 | 2.79 | 1.68 | 8.30 | 8.80 | 3.16 | 0.89 |
| [1]Saturated fatty acids (SFAs) | 35.20 | 36.61 | 47.09 | 20.01 | 49.31 | 49.55 | 55.45 | 21.03 |
| [2]Unsaturated fatty acids (UFAs) | 59.00 | 58.44 | 50.11 | 79.08 | 41.90 | 41.14 | 41.13 | 78.01 |
| Monounsaturated fatty acids (MUFAs) | 43.25 | 45.72 | 39.83 | 34.44 | 32.92 | 32.55 | 34.44 | 32.50 |
| Polyunsaturated fatty acids (PUFAs) | 15.75 | 12.72 | 10.28 | 44.64 | 8.98 | 8.59 | 6.69 | 45.51 |
| Omega 3 (ω3) | 1.08 | 0.67 | 0.69 | 15.32 | 4.23 | 4.13 | 2.06 | 17.10 |
| Omega 6 (ω6) | 14.67 | 12.05 | 9.59 | 29.32 | 4.75 | 4.46 | 4.63 | 28.41 |
| Omega 9 (ω9) | 34.57 | 36.34 | 34.58 | 31.9 | 16.09 | 15.73 | 25.97 | 29.32 |
| Ratio (ω6/ω3) | 13.58 | 17.98 | 13.90 | 1.91 | 1.12 | 1.08 | 2.25 | 1.66 |
| UFAs/SFAs ratio | 1.68 | 1.60 | 1.06 | 3.95 | 0.85 | 0.83 | 0.74 | 3.71 |

*Other fatty acids found in minimal amounts when evaluated individually.

## Discussion

Fillet yield and by-products generated during filleting process are aspects of important economic value, not only for the processing of fish, although throughout the entire aquaculture production chain. There are several factors that interfere and influence the fish fillet yield, such as its species and consequently its morphological characteristics, such as anatomical shape, size, head/body ratio, body mass, chemical composition of the animal and sex, in addition to technological factors such as the filleting method, the degree of mechanization of the

processing unit and qualification of the filleting operator [24, 25]. During the processing of Nile tilapia and croaker, several of these factors were present, because the fish are of different species and different body weights were obtained, 0.75 kg for Nile tilapia and 2.82 kg for croaker (**Table 1**). In addition, the fish were obtained and processed in different regions and methodologies.

Some studies have evaluated yields of Nile tilapia fillets, both at industrial and experimental scales, applying different filleting methods [26]. After filleting the tilapia in this study, the average fillet yield was 32.6% (**Table 1**), a value close to that obtained by Peterman & Phelps [27], which was 31%, while other authors obtained average fillet yields smaller or larger than those of this study, as observed by Morais et al. [28] which was 28.4% for Nile tilapia, with body masses ranging 0.751 to 1.0 kg and Leonhardt et al. [29] who observed an average fillet yield of 42% for the same fish species. However, croaker showed an average fillet yield of 56.6%, which is higher than that of Nile tilapia and other studies such as the one conducted by Leira et al. [30], whose mentioned yield value reached 42.2% (fish with an average body mass of 1.278 kg). This showes that for this species there is a variation in the yield of its fillet, according to body mass of the fish at the time of slaughter. However, other factors may be involved in this difference in income. Since the croaker from fishing in the Amapá state has viscera removed while still inside the vessels on the high seas, to reduce the possibility of deterioration of the fillet, this fact affects overestimating the fillet yield of this species.

Muscle tissue *in natura* of fish has a high moisture, regardless of the species evaluated, however, there is also a variation between species. Muscle of *Mugil cephallus*, for example, has 78.4% of moisture, the true sardine (*Sardinella brasiliense*) has a content 72% [31], values lower than those found for Nile tilapia 80.24% and croaker 82.87% in this study (**Table 1**). The lowest moisture contents found by Viana et al. [31] are explained by the presence in greater amounts of other nutrients, mainly crude protein, which ranged from 17 to 23.4% between mullet and true sardines. In this study, the moisture content was much higher, although the protein content was low for croaker fillet 14.21%, consequently higher moisture content 82.87%.

When comparing the values found for flour made with Nile tilapia backbone in this study, with those demonstrated by Petenuci et al. [32], it is observed that the moisture and lipid contents were much lower, 3.53 and 4.28%, respectively (**Table 2**). This makes it clear that the fish-flour production methodology directly interferes with the quality of the final product, since the methodology used by Petenuci et al. [32] did not perform the pressing step, which was carried out in current study and is responsible for removing much of the moisture and lipids during the process of obtaining the flour. In this process, a force of around 10 tons is applied, providing greater leaching of the liquid part present in water and in natural lipid of the backbone. Flours produced using Nile tilapia and croaker heads as raw material had lower average levels of protein 39.86%, when compared to average protein value of the flours obtained from backbones of these fish species 49 .58% (**Table 2**). The value 39.86% observed for protein content in head flours are very close to what was reported by Justen et al. [33], which was 38.41% for the same type of Nile tilapia head flour.

Casaretto et al. [34] studied with Nile tilapia MSM protein concentrate and the authors reported much higher protein levels and 62.39% than the levels found in this study, although this result was due to fact that the raw material used by authors was MSM that did not has pimples. For lipid content, salted flour showed 8.06%, a value significantly higher than that of sweet flour 3.82%, this effect can explained by inclusion in the formulation of salted sesame flour, sesame oil and flaxseed, which are oleaginous foods. The inclusion of these ingredients in the flour provided an increase of 49.13% of lipids when compared to standard flour of the backbone (4.10%).

Essential amino acids are those that the body cannot synthesize, however they are necessary for its functioning, these nutrients are crucial for the quality of the protein fraction of a food [35]. Among the essential amino acids, lysine is extremely important, as it must be consumed in diet of practically all animals, and Nile tilapia backbone flour expressed a content 5.31% for this amino acid, while the backbone flour of croaker showed a lower content 3.12% (**Table 3**). According to WHO [36] the amino acids that meet the nutritional requirements recommended for adults are 1.6g 100g$^{-1}$ of histidine, 1.3g 100g$^{-1}$ of isoleucine, 1.9g 100g$^{-1}$ of leucine, 1.6g 100g$^{-1}$ of lysine, 1.7g 100g$^{-1}$ of methionine+cystine, 0.9g 100g$^{-1}$ of threonine, 0.5g 100g$^{-1}$ of tryptophan and 1.3g 100g$^{-1}$ of valine. Therefore, comparing the two flours made with fish bones of the two species studied, it is noted that Nile tilapia better meets the nutritional requirements recommended for adults. Only histidine was below the recommended level, while croaker backbone flour were histidine and tryptophan, which were limiting in the two flours prepared. Therefore, between the two flours, the one with the best nutritional quality is that of backbone.

During the fishflour production process, an important step is dehydration, as water is lost to environment, reducing moisture and water activity and consequently increasing its storage time or shelf life. However, this equipment requires a significant amount of electrical energy and the decrease in dehydration time, leads to energy and financial savings in the manufacture of this type of product.

Some studies have agreed in their methodologies to use forced-air ovens with a temperature of 60˚C, within a 24 hours period [10, 22, 37]. The optimal time for dehydration of Nile tilapia flour was 15 hours and 20 minutes and for croaker flour was 15 hours and 42 minutes, while several authors [10, 22, 37, 38], reported that dehydration period for flours should 24 hours and a temperature 60˚C was used during the process. However, it is possible to reduce the dehydration period of the flours by around 8 hours. Perhaps this time reduction is associated with the moisture and initial granulometry of the raw material, temperature used in the dehydration process, because for this study of the two species a temperature 90˚C was used this stage is completed at that moment, providing a reduction in the production cost of the same, in relation to energy expenses. Another important factor for validating the optimal dehydration time was the moisture content obtained for tilapia and croaker backbone flour of 3.53 and 3.95%, respectively (**Table 2**), with values very close to ($P < 0.05$). These levels were also lower than those established by RIISPOA [39], which describes that these types of fish products must not contain more than 12% moisture in their composition.

The aW values of sweet flours were lower than salted ones. This fact is due to difference in methodology used for production of flour, where the sweet flour, after the elaboration process, was again placed in an oven at 60˚C, for 24 hours. Costa et al. [40] when producing flour from Nile tilapia minced our mechanically separated meat (MSM) observed different water activities, when salting is applied or not in storage of material, before the preparation of the flour, since the salted materials obtained values that ranged 0.5256 to 0.8434 for this type of the flour. These values were higher than the 0.19 indicated by this study. However, for MSM flours without salting, during the storage process, the same authors reported values between 0.113 and 0.432 for the same parameter (aW). Values of aW lower than 0.60 are important as they guarantee better microbiological stability of the product [33].

Fish meat has a pH close to neutrality, as well as high aW, which can cause the growth of some undesirable microorganisms. With the objective of using the carcasses and heads for human consumption, an alternative would to reduce the water content, facilitating its application in several products and ensuring a longer shelf life of the product, producing the flours. Despite the values close to neutrality found in current study, the microbial quality of Nile tilapia and croaker flours, made from fish backbone and heads, was adequate. These results

showes that for production of seasoned flours a good hygienic-sanitary criterion was used, and they can used for inclusion in food products without any problem, as the criteria of Good Manufacturing Practices for food products were followed.

According to Costa et al. [40] when producing flour from Nile tilapia MSM, using the pre-treatment of 60˚C, the closest to this study, the same authors obtained in the colorimetric analysis the values 51.68 for luminosity, 3.39 for a* chromaticity and 13.19 for b* chromaticity. Thus, the luminosity was lower than all the values obtained in Nile tilapia and croaker flour using the backbones and heads of the animals, in addition, the two chromaticities obtained were higher, indicating that the flour from Costa et al. [40], in addition to being darker, it had more pigments, which tended to increase the a* and b* chromaticity, which, in this difference, is due to way the authors managed cleaning of the backbones before obtaining MSM, possibly leaving traces of clotted blood and residual parts of fish visceral, which result in darker flours. There is also the issue of washing MSM for flour production, as when washing or several washing cycles are used, there is greater leaching of nutrients, consequently making the flour lighter [8, 41].

Souza et al. [42] observed that flours obtained from Nile tilapia MSM expressed lower luminosity 55.32 than head flour 62.49, with the backbone flour 77.19 being the flour with greater luminosity. The authors state that the high lipid content 13.15% of this flour influenced the reduction of luminosity, leaving the darker compared to head. These authors reported that head and (carcass—backbone) flours had higher values of chroma a*, tending to red, and these were very similar to those obtained in this study (**Table 4**), with chromium b*, showes lower values. The head flour had the lowest chroma b* value (4.13) then the carcass flour 5.29 and MSM the highest value 9.51, but all the chroma b* values obtained by Souza et al. [42] were lower than those obtained for flours in this experiment.

Determination of flours granulometry is very important, since its main purpose is to add to food products for the purpose of enrichment. In this way, the granulometry, being of a smaller size, will be easier for incorporation into these food products, because it physically characterizes the ingredient produced, as flours with smaller granulometries are easier for incorporation into food products [41, 43].

The milling step is also decisive for size of the DGM, as factors such as calibration of device or the use or not of sieves are responsible for the different values of this parameter. In this study, at time of flour milling, sieves were not used, providing different granulometries between treatments. However, even with different granulometries, all treatments were classified as fine-grained and according to Vukmirović et al. [19] this classification is received when an ingredient has an average geometric diameter of less than 0.60 mm. Thus, the different flours prepared were classified as fine despite statistical differences observed between them.

Nile tilapia head flour showed the highest 0.56 and generally the lipid content is responsible for agglutination of particles, taking into account that even with no interaction between four treatments of the elaborated flours regarding their lipid content. Petenuci et al. [32] prepared Nile tilapia carcass flour that demonstrated 18.3% of ash, while Nile tilapia of this study expressed 28.32% for the same type of raw material (backbone) and when seasoned (salted) it was par 17.82%. Therefore, the fish species, flour production methodology and the flavoring technique affect the amount of mineral matter (ash) and consequently the minerals (calcium and phosphorus analyzed). Croaker flours (salted and sweet) had higher levels of calcium and phosphorus compared to Nile tilapia (salted and sweet), with sweet croaker having the highest levels of these minerals and salted tilapia the lowest.

The pH values found in flours was considered neutral (**Table 5**). However, when temperadas, there was a significant pH variation, with salted flours showes a reduction in pH 6.37, being a more acidic product, while sweet flours had a higher pH 7.71, a more alkaline product. Chambó et al. [44] reported similar values for their flours, where for carcass the pH was 7.33 and for MSM

7.13. The inclusion of seasonings provided a greater amount of hydrogen ions (H$^+$), increased acidity in salted flours, that is, reducing the pH of the product. If comparing these pH results of these seasoned flours with the standard backbone flours (**Table 2**), it is noticed that what really influenced the change in pH of the flours was the flavoring and not the species of fish used. In **Table 2**, it is observed that the flours had a pH value that ranged 7.0 to 7.08.

When dealing with fish-based products, it is important to analyze the fatty acids found, as they are rich sources of omega 3, 6 and 9. The differences between the concentration of fatty acids in the flours can explained by leaching process of some nutrients that can occur during production of the flours. And the difference in values mainly of Oleic and Linoleic acids in the flours of two species with addition of salted flavor is due to ingredient added in the flours, such as flaxseed, which is rich in polyunsaturated fatty acids [45], being that its addition contributes to increase of polyunsaturated acids in salted flours, where salted Nile tilapia flour has 44.64 and salted croaker 45.51 of polyunsaturated fatty acids.

## Conclusions

Nile tilapia has a lower fillet yield, although with a high protein content and low lipid content compared to croaker fillet. Comparing the flours made from filleting by-products, backbone flour has better nutritional quality, with tilapia being superior to that of croaker, especially in terms of protein and amino acids. With seasoned flours (salted and sweet) there is a reduction in the protein content, therefore, the standard flour (without seasoned) is more suitable for inclusion in food products, with purpose of greater enrichment in protein and minerals. However, when it comes to seasoned flour, the sweets regardless of the species used (Nile tilapia or croaker) have higher levels of protein, however, when it comes to lipid fraction, salted (seasoned) flours are more suitable for consumption, as they have a higher amount of omega 3 series fatty acids. All flours were within microbiological standards and fit for consumption.

It is concluded that the methodologies used for elaboration of flours are propitious to generate a good product, of nutritional and microbiological quality, which can added to other foods, with the purpose of enriching protein, minerals, fatty acids as well as reducing the possibility of environmental impact and adding value to production chain of these fish species.

## Author Contributions

**Conceptualization:** Stefane Santos Corrêa, Gislaine Gonçalves Oliveira, Melina Coradini Franco, Eliane Gasparino, Andresa Carla Feihrmann, Simone Siemer, Jerônimo Vieira Dantas Filho, Jucilene Cavali, Sandro de Vargas Schons, Maria Luiza Rodrigues de Souza.

**Data curation:** Stefane Santos Corrêa, Gislaine Gonçalves Oliveira, Melina Coradini Franco, Eliane Gasparino, Andresa Carla Feihrmann, Simone Siemer, Jerônimo Vieira Dantas Filho, Jucilene Cavali, Sandro de Vargas Schons, Maria Luiza Rodrigues de Souza.

**Formal analysis:** Stefane Santos Corrêa, Gislaine Gonçalves Oliveira, Melina Coradini Franco, Eliane Gasparino, Andresa Carla Feihrmann, Simone Siemer, Jerônimo Vieira Dantas Filho, Jucilene Cavali, Sandro de Vargas Schons, Maria Luiza Rodrigues de Souza.

**Funding acquisition:** Stefane Santos Corrêa, Gislaine Gonçalves Oliveira, Melina Coradini Franco, Eliane Gasparino, Andresa Carla Feihrmann, Simone Siemer, Jerônimo Vieira Dantas Filho, Jucilene Cavali, Sandro de Vargas Schons, Maria Luiza Rodrigues de Souza.

**Investigation:** Stefane Santos Corrêa, Gislaine Gonçalves Oliveira, Melina Coradini Franco, Eliane Gasparino, Andresa Carla Feihrmann, Simone Siemer, Jerônimo Vieira Dantas Filho, Jucilene Cavali, Sandro de Vargas Schons, Maria Luiza Rodrigues de Souza.

**Methodology:** Stefane Santos Corrêa, Gislaine Gonçalves Oliveira, Melina Coradini Franco, Eliane Gasparino, Andresa Carla Feihrmann, Simone Siemer, Jerônimo Vieira Dantas Filho, Jucilene Cavali, Sandro de Vargas Schons, Maria Luiza Rodrigues de Souza.

**Project administration:** Stefane Santos Corrêa, Gislaine Gonçalves Oliveira, Melina Coradini Franco, Eliane Gasparino, Andresa Carla Feihrmann, Simone Siemer, Jerônimo Vieira Dantas Filho, Jucilene Cavali, Sandro de Vargas Schons, Maria Luiza Rodrigues de Souza.

**Resources:** Stefane Santos Corrêa, Gislaine Gonçalves Oliveira, Melina Coradini Franco, Eliane Gasparino, Andresa Carla Feihrmann, Simone Siemer, Jerônimo Vieira Dantas Filho, Jucilene Cavali, Sandro de Vargas Schons, Maria Luiza Rodrigues de Souza.

**Software:** Stefane Santos Corrêa, Gislaine Gonçalves Oliveira, Melina Coradini Franco, Eliane Gasparino, Andresa Carla Feihrmann, Simone Siemer, Jerônimo Vieira Dantas Filho, Jucilene Cavali, Sandro de Vargas Schons, Maria Luiza Rodrigues de Souza.

**Supervision:** Stefane Santos Corrêa, Gislaine Gonçalves Oliveira, Melina Coradini Franco, Eliane Gasparino, Andresa Carla Feihrmann, Simone Siemer, Jerônimo Vieira Dantas Filho, Jucilene Cavali, Sandro de Vargas Schons, Maria Luiza Rodrigues de Souza.

**Validation:** Stefane Santos Corrêa, Gislaine Gonçalves Oliveira, Melina Coradini Franco, Eliane Gasparino, Andresa Carla Feihrmann, Simone Siemer, Jerônimo Vieira Dantas Filho, Jucilene Cavali, Sandro de Vargas Schons, Maria Luiza Rodrigues de Souza.

**Visualization:** Stefane Santos Corrêa, Gislaine Gonçalves Oliveira, Melina Coradini Franco, Eliane Gasparino, Andresa Carla Feihrmann, Simone Siemer, Jerônimo Vieira Dantas Filho, Jucilene Cavali, Sandro de Vargas Schons, Maria Luiza Rodrigues de Souza.

**Writing – original draft:** Stefane Santos Corrêa, Gislaine Gonçalves Oliveira, Melina Coradini Franco, Eliane Gasparino, Andresa Carla Feihrmann, Simone Siemer, Jerônimo Vieira Dantas Filho, Jucilene Cavali, Sandro de Vargas Schons, Maria Luiza Rodrigues de Souza.

**Writing – review & editing:** Stefane Santos Corrêa, Gislaine Gonçalves Oliveira, Melina Coradini Franco, Eliane Gasparino, Andresa Carla Feihrmann, Simone Siemer, Jerônimo Vieira Dantas Filho, Jucilene Cavali, Sandro de Vargas Schons, Maria Luiza Rodrigues de Souza.

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
