## [Decision Letter · Decision Letter 0]

12 Sep 2022

PONE-D-22-17293Quality of Oreochromis niloticus and Cynoscion virescens fillets and their by-products in flours make for inclusion in instant food productsPLOS ONE

Dear Dr. Gonçalves Oliveira,

Thank you for submitting your manuscript to PLOS ONE. After careful consideration, we feel that it has merit but does not fully meet PLOS ONE’s publication criteria as it currently stands. Therefore, we invite you to submit a revised version of the manuscript that addresses the points raised during the review process.

We look forward to receiving your revised manuscript.

Kind regards,

Ashokkumar Veeramuthu

Academic Editor

PLOS ONE

Journal Requirements:

CAPES/Brazil through Programa Nacional de Cooperação Acadêmica na Amazônia (PROCAD/AM). Project approved 2018/L.1, in the postgraduate line in Biodiversity, production and animal health. All authors were equally awarded.

NO authors have competing interests

Additional Editor Comments:

Editor Comments:

1. The authors must highlight the significance of this work by addressing the current problem statement, novelty, clear study objectives, and future direction in the introduction section.

2. In some sections, the discussion part is weak, please provide an in-depth discussion by comparing the most recent literature survey on this particular area. Also, when you make a comparison with reported literature, please consider referring to the most recent works (2019 – 2023).

3. This manuscript contains some technical and grammatical mistakes; the authors must go for a thorough technical and language check.

4. Please improve the quality of all the figure and tables, the present form did not meet the journal standard.

Reviewers' comments:

Reviewer's Responses to Questions

**Comments to the Author**

1. Is the manuscript technically sound, and do the data support the conclusions?

Reviewer #1: Yes

Reviewer #2: Yes

2. Has the statistical analysis been performed appropriately and rigorously? 

Reviewer #1: Yes

Reviewer #2: Yes

3. Have the authors made all data underlying the findings in their manuscript fully available?

Reviewer #1: Yes

Reviewer #2: Yes

4. Is the manuscript presented in an intelligible fashion and written in standard English?

Reviewer #1: Yes

Reviewer #2: Yes

5. Review Comments to the Author

Reviewer #1: The work describes ways of using fish filleting by-products. Studies in this line of research are very important, especially in Brazil, where most of the waste is still underused, and often improperly discarded. The work demonstrates the possibility of using filleting residues in the preparation of flour for human consumption. It is a robust work with a potential positive impact for the area of technology for fish products. However, some points must be improved so that the article can be published.

Abstract:

-The methodology for obtaining “Flavored flours” is not demonstrated

Introduction

-Line 64: studies that indicate the maximum size of the species must be cited and referenced

-Line 66: the phrase “Most saltwater caught fish are imported.” has no relation to the context and should be excluded

-It is not clear in the introduction the justification for the study of the species Cynoscion virescens.

-The introduction does not defend the importance of studies on the quality and yield of fillets of the species studied. In particular for Nile tilapia, several studies have already addressed this issue (especially on centesimal composition, quality analysis performed in the study). Thus, the introduction must contemplate the novelty that the article brings in this field of study, in relation to articles already published.

- Greater theoretical reference could be added on the production of fishmeal for human consumption, and the reason for developing flavored flours.

Material and methods

-Describe the origin of the fish used in the study

-Describe method of stunning and slaughtering animals

-Model, brand and country of origin data must be entered on all mentioned equipment.

-line 117: review information: oven drying at 90°? This temperature seems excessive to me

-line 122: which products were developed with the flours? This information is new in the article

-line 125: We can only call it “flour flavored” if the authors carried out, for example, a chromatographic analysis to identify and quantify the volatile aromatic components present in the sample. Proving that in fact the flour has flavoring power. Empirically, we cannot classify anything. I suggest changing the title and revising the terms “flavored/flavored” throughout the article.

-line 125: what is the final objective of these sweet and savory flavored flours? The rationale for these products should be shown in the introduction.

-line 126: why were the flavored flours produced only with backbone flour? Why not also use head flour?

-line 129: wouldn't it be monosodium glutamate?

-line 135: after removing from the fire, what was the next methodological step to obtain the flour?

- lines 128 to 139: were these ingredients used in these proportions according to the previous work?

-line 140: the Arapaima gigas (paiche) carcass flour?

-line 146 onwards: could have subtitles for the analyzes performed, for better organization of the text

-line 150: how many fillet samples were analyzed? N=10?

-line 174: mention the ABNT standards

-lines 182-183: This resolution has already been revoked. Adapt the target microorganisms and their respective limits according to RDC 331/2019 and IN 60/2019, both from ANVISA.

-Statistical design: what treatment of data was given for the analysis of fatty acids and amino acids?

Results

-The results section in general is confusing. Results could be grouped into subheadings (at least in relation to Test 1 and Test 2).

-The presentation of the results is confused with the discussion. See below:

-Line 198-199: this sentence belongs to Discussion

-lines 211-212: this sentence belongs to Discussion

-lines 221-222: this sentence belongs to Discussion

-lines 223-226: these sentences belong to Discussion

-lines 246-251: these sentences belong to Discussion

-lines 255-258: these sentences belong to Discussion

-lines 260-266: these sentences belong to Discussion

-lines 276-279: these sentences belong to Discussion

-lines 291-293: this sentence belongs to Discussion

-lines 297-299: this sentence belongs to Discussion

-lines 302-306: these sentences belong to Discussion

-lines 319-323: these sentences belong to Discussion

-lines 327-328: these sentences belong to Discussion

-Lines 351-353: This information about the croaker (which does not appear in the methodology), makes it impossible to calculate the fillet yield and the comparison with the tilapia results. If the authors do not know the weight of the whole croaker, there is no way to specify the fillet yield (56.6% is a very high fillet yield), but if it was calculated based on the eviscerated fish, it does not represent the actual fillet yield. . So I suggest removing the yield results from the article.

-Table 2: Standardize test letters of averages (put all superscripts). When there is no significant effect (P>0.05), it is not necessary to add letters after the means (add letters only when the effect was significant). Please review this in all tables. In addition, I could present Table 2 in the same way as Table 4 was presented, it facilitates the understanding of the effects.

-Table 3: Why was it chosen to perform an analysis of the amino acid profile only for backbone flours?

-Table 5 could be presented as Table 4

Discussion

-Line 394: The authors report that a temperature of 90°C was used for the dehydration of the flours. However, it is known that high temperatures during drying lead to several irreversible biological or chemical reactions, as well as structural, physical and mechanical changes. Several works that developed fish meal used milder temperatures. What led the authors to use 90°C?

-Line 398: Where is “RIISPOA [34]”, the correct one is [24]

-Lines 400-405: Why is Aw important in this type of product? To describe.

-Lines 406-408: it's confused

-The discussion needs to be reformulated according to sentences that belong to this section, but are in the results.

-Lines 428-430: The statement is unnecessary, since the work did not use MSM as raw material

-Lines 461-462: is confused

- Discussion about the fatty acid (Table 6) and amino acid contents of the flours (Table 3) should be inserted.

Conclusions

-Line 467: Review the conclusion about the lowest fillet yield, based on the comment made about this analysis for croaker.

-Lines 480-481: It is not possible to conclude based on the acceptability of the products, since sensory analysis was not performed in this work.

Reviewer #2: In this paper, the authors discussed the the elaboration of flours to generate a good product, of nutritional and microbiological quality, which can be added to other foods, with the purpose of enriching protein, minerals, fatty acids and providing an increase in the acceptability in food products, as well as reducing the possibility of environmental impact and adding value to production chain of these fish species.

This paper contains new and enough interesting results after taking into account the following comments

1) The English should be improved

2) In the introduction section, the authors write line 76 "" With the need for a more conscious and environmentally correct use of by-products and waste from fish filleting, this fact makes food industries seek viable solutions for the recovery of by-products and waste, seeking new alternatives for their use in a more efficient way more efficient and healthier for consumers of fish products [.......].

The following references may be cited

Wenya Tian, Design and Implementation of Web-Based Food Regulatory Information Resources Management Platform

Applied Mathematics & Information Sciences, Volume 05, No. 5-2S PP: 105S-111S (2011)

N. Subramanian, K. Saravanan,

Regime Classification of Geldart B Food Particles in Circulating Fluidized Bed

Applied Mathematics & Information Sciences, Volume 13, No. 4 PP: 589-594 (2019) doi:10.18576/amis/130410

Najat O. A. Al-Salahi, Magda M. S. Saleh, Elham Y. Hashem,

Utility of Spectrophotometry for Novel Quantitation of Sudan Orange G in some Commercial Food Products

Journal of Pharmaceutical and Applied Chemistry, Vol. 5, No. 3 PP: 117-129 (2019)

3) More discussion on For preparation of sweet flour should be added

4) In the Statistical design section, the authors needs to add in a clear way some discussion on the used sample.

5) The authors write "The pH of fish provides conditions that stimulate microbial multiplication, as its meat has a pH

close to neutrality, in addition to high water activity " This is not clear ??

6) In the references list the same format should be used in each reference

6. PLOS authors have the option to publish the peer review history of their article (what does this mean?). If published, this will include your full peer review and any attached files.

Reviewer #1: No

Reviewer #2: No

---

## [Author Response · Author response to Decision Letter 0]

28 Oct 2022

Justifications

- lines 128 to 139: were these ingredients used in these proportions as in the previous work?

Not. The ingredients and quantities were proposed by the researchers.

-Table 3: Why was it chosen to perform the analysis of the amino acid profile only for backbone flours?

Due to the high cost, it would not possible to make amino acids from all flours, so we chose to present the one with the best result in terms of protein content.

-Line 394: The authors report that a temperature of 90°C was used for the dehydration of the flours. However, it is known that high temperatures during drying lead to several irreversible biological or chemical reactions, in addition to structural, physical and mechanical changes. Several works that developed fishmeal used milder temperatures. What led the authors to use 90°C?

As the carcasses and heads were cooked for 60 minutes, for the preparation of the flours, it was decided to use a higher temperature in order to reduce the dehydration time and to reduce the chance of any contamination during the milling process of the raw materials. With dehydration at a higher temperature, it can reduce or eliminate the possibility of contamination by microorganisms., in addition to comparing the results obtained with the literature? Comparing the results obtained, it can observed that many parameters were superior to those in the literature.

All other adjustment requests were accepted/accomplished.

Competing interests: The authors have declared that no competing interests exist.

Data Availability: All relevant data is in the paper and its Supporting information files.

Funding: This study received financial support from CAPES/Brasil through Programa Nacional de Cooperação Acadêmica na Amazônia - PROCAD-AM (UNIR/UFAC/USP), for the granting of a postdoctoral scholarship to Jerônimo Vieira Dantas Filho.

Competing interests: The authors have declared that no competing interests exist.

---

## [Decision Letter · Decision Letter 1]

6 Dec 2022

Quality of Oreochromis niloticus and Cynoscion virescens fillets and their by-products in flours make for inclusion in instant food products

PONE-D-22-17293R1

Dear Dr. Gonçalves Oliveira,

We’re pleased to inform you that your manuscript has been judged scientifically suitable for publication and will be formally accepted for publication once it meets all outstanding technical requirements.

Kind regards,

Ashokkumar Veeramuthu

Academic Editor

PLOS ONE

Additional Editor Comments (optional):

Reviewers' comments:

Reviewer's Responses to Questions

**Comments to the Author**

1. If the authors have adequately addressed your comments raised in a previous round of review and you feel that this manuscript is now acceptable for publication, you may indicate that here to bypass the “Comments to the Author” section, enter your conflict of interest statement in the “Confidential to Editor” section, and submit your "Accept" recommendation.

Reviewer #1: All comments have been addressed

Reviewer #2: All comments have been addressed

2. Is the manuscript technically sound, and do the data support the conclusions?

Reviewer #1: Yes

Reviewer #2: Yes

3. Has the statistical analysis been performed appropriately and rigorously? 

Reviewer #1: Yes

Reviewer #2: Yes

4. Have the authors made all data underlying the findings in their manuscript fully available?

Reviewer #1: Yes

Reviewer #2: Yes

5. Is the manuscript presented in an intelligible fashion and written in standard English?

Reviewer #1: Yes

Reviewer #2: Yes

6. Review Comments to the Author

Reviewer #1: (No Response)

Reviewer #2: This version can be accepted in its present form. Most of the comments have been implemented

This version can be accepted in its present form. Most of the comments have been implemented

7. PLOS authors have the option to publish the peer review history of their article (what does this mean?). If published, this will include your full peer review and any attached files.

Reviewer #1: No

Reviewer #2: No

---

## [Editor Report · Acceptance letter]

13 Dec 2022

PONE-D-22-17293R1 

Quality of *Oreochromis niloticus* and *Cynoscion virescens* fillets and their by-products in flours make for inclusion in instant food products 

Dear Dr. Gonçalves Oliveira:

I'm pleased to inform you that your manuscript has been deemed suitable for publication in PLOS ONE. Congratulations! Your manuscript is now with our production department. 

Kind regards, 

on behalf of

Dr. Ashokkumar Veeramuthu 

Academic Editor

PLOS ONE